# Rapid Screening of Microalgae as Potential Sources of Natural Antioxidants

**DOI:** 10.3390/foods12142652

**Published:** 2023-07-10

**Authors:** Na Wang, Haiwei Pei, Wenzhou Xiang, Tao Li, Shengjie Lin, Jiayi Wu, Zishuo Chen, Houbo Wu, Chuanmao Li, Hualian Wu

**Affiliations:** 1CAS Key Laboratory of Tropical Marine Bio-Resources and Ecology, Guangdong Key Laboratory of Marine Materia Medica, RNAM Center for Marine Microbiology, Institution of South China Sea Ecology and Environmental Engineering, South China Sea Institute of Oceanology, Chinese Academy of Sciences, Guangzhou 510301, China; nawang@scsio.ac.cn (N.W.); haiweipei@scsio.ac.cn (H.P.); xwz@scsio.ac.cn (W.X.); taoli@scsio.ac.cn (T.L.); kayeewu@scsio.ac.cn (J.W.); 18390943716@163.com (Z.C.); wuhoubo@scsio.ac.cn (H.W.); 2University of Chinese Academy of Sciences, Beijing 100049, China; 3Southern Marine Science and Engineering Guangdong Laboratory (Guangzhou), No. 1119, Haibin Road, Nansha District, Guangzhou 511458, China; 4Guangzhou Keneng Cosmetic Scientific Research Co., Ltd., Guanghzou 510800, China; linsj@danzi.cn (S.L.); lichm@danzi.cn (C.L.)

**Keywords:** microalgae, antioxidant activity, carotenoid content, phenolic content, weighted scoring system

## Abstract

In order to rapidly screen microalgae species as feedstocks for antioxidants, extracts were obtained from 16 microalgae strains (under 11 genera, 7 classes) using two methods: a one-step extraction with ethanol/water and a three-step fractionating procedure using hexane, ethylacetate, and water successively. Measuring the total phenol content (TPC), total carotenoid content (TCC), and antioxidant activity of the extracts, indicating TPC and TCC, played an important role in determining the antioxidant activity of the microalgae. A weighted scoring system was used to evaluate the antioxidant activity, and the scores of microalgal samples from two extraction methods were calculated using the same system. Among the investigated microalgae, *Euglena gracilis* SCSIO-46781 had the highest antioxidant score, contributing to high TPC and TCC, followed by *Arthrospira platensis* SCSIO-44012, *Nannochloropsis* sp. SCSIO-45224, *Phaeodactylum tricornutum* SCSIO-45120, and *Nannochloropsis* sp. SCSIO-45006, respectively. Additionally, the above-mentioned five strains are currently being applied in commercial production, indicating this system could be effective not only for screening microalgal antioxidants, but also for screening microalgal species/strains with strong adaptation to environmental stress, which is a critical trait for their commercial cultivation.

## 1. Introduction

A large number of reactive oxygen species (ROS) are produced during human activities. However, excess ROS causes a variety of diseases, such as diabetes, cancer, cardiovascular disease, Alzheimer’s disease, etc. [1]. Antioxidants are biological macromolecules that can inhibit or reduce ROS and oxidative stress molecules [2]. With the increasing complexity of the human living environment, the antioxidant endogenous machinery in humans, although highly efficient, has been unable to meet the normal needs of the human body to counteract the development or harmful effects of ROS. Therefore, a supplement of exogenous antioxidant molecules is required [3]. Furthermore, synthetic antioxidants, such as butylhydroxytoluene (BHT) and butylhydroxyanisole (BHA), have been reported to cause safety concerns, which can lead to mutagenic changes and carcinogenic effects [4]. Therefore, it has become a global trend to look for new, low-cost, powerful, green, and safe exogenous natural antioxidants to replace chemical antioxidants.

Exogenous antioxidants are a large group of molecules, including carotenoids, polyphenols, vitamins and their derivatives, and antioxidant minerals [5]. At present, most of the natural antioxidants on the market come from land plants, and antioxidants extracted from microalgae are not common in the market. However, studies on the antioxidant activity of microalgae have found that some microalgae contain substances with high antioxidant capacity and have the advantage of being used as natural antioxidants. Additionally, microalgae are rich in pigments, phenols, polysaccharides, proteins, essential fatty acids, vitamins, mineral oxides, and other high-value biologically active nutrients, which can serve as potential sources of natural antioxidants and are widely used in nutritional food and pharmaceutical and cosmetical industries [6]. For instance, a blend of inorganic nanoparticles and natural antioxidant chemicals of microalgae was popular in active food packaging due to their robust antibacterial, antioxidant, UV barrier, oxygen removal, and low environmental impact properties [7,8]. Additionally, the food industry is applying whole microalgal biomass or extracted purified compounds as novel ingredients for the formulation of food products such as baked goods, pasta, noodles, plant-based milk, soups, and many others [9]. Ayna et al., (2020) [10] reported that beta-tocopherol and alpha-carotene from microalgae served as oxidants in prostate cancer by reducing cell viability and increasing ROS production and lipid peroxidation. Foo et al., (2021) [11] found that carotenoids can be used as natural antioxidants in anti-aging skin care products in the cosmetics industry, in addition to being incorporated as a natural pigment to color cosmetics.

In particular, carotenoids are the best-studied antioxidative chemicals, owing to their bioactivity and potential advantages for human health [12]. Microalgae contain various carotenoids, such as β-carotene, astaxanthin, fucoxanthin, lutein, kerataxanthin, zeaxanthin, and lycopene, which are natural colorants and fat-soluble plant pigments with significant antioxidant activity [13]. It has been reported that the β-carotene content of *Dunaliella salina* could comprise up to 10.0% of its dry weight [14] and the diatom *Phaeodactylum tricornutum* accumulated fucoxanthin with a yield of 8.32 mg L^−1^ [15]. In addition, β-carotene, zeaxanthin, and β-cryptoxanthin are the main carotenoids in *Spirulina platensis* [16]. *Chlorella*, *Haematococcus*, *Dunaliella*, and *Chlamydomonas*, which are essential sources of carotenoid extraction [17]. Furthermore, carotenoid molecules could act as lipid radical scavengers and singlet oxygen quenchers to protect against oxidant stress [18]. β-carotene and lycopene exhibit higher antioxidant activity than other carotenoids [19], and astaxanthin extracted from *Haematococcus pluvialis* has been widely used as a highly effective natural antioxidant [20].

Literature has reported that phenols, transported by single electrons and hydrogen atoms, are the main source of commonly used antioxidants [21]. Marine microalgae contain a variety of polyphenols, such as green tannins, which have antioxidant properties [22]. Phenolic compounds are involved in the process of inhibiting natural stresses. Although many phenolic compounds have been observed in *Chlorella pyrenoidosa* or *S. platensis*, the research on these compounds is not sufficient [23]. Additionally, it is generally believed that carotenoids and phenolic compounds are the main contributors to antioxidant activity in microalgae, but knowledge about these compounds and their antioxidant properties is scarce [24,25]. Due to the diversity of microalgae species and the variety of extraction methods used for microalgae biomass, it is complicated to establish the correlation between the content of compounds extracted from microalgae and their antioxidant capacity.

Additionally, there are abundant algae resources in the world, and all of which may be potential candidates for development in the algae farming industry [26,27]. *Chlorella*, *Spirulina*, *Porphyridium*, *Dunaliella*, and diatom *Skeletonema marinoi* have been successively proposed by different researchers [12,28,29,30,31]. However, so far, few algae strains have been found in large-scale cultivation as natural antioxidants. It is undoubtedly necessary to rapidly screen out cultivable algae with high antioxidant activity in microalgae industry development.

In light of this, 16 strains of microalgae were extracted using two different solid–liquid extraction methods. The contents of carotenoids, phenols, and antioxidant activity were assayed. Then, a weighted scoring system was used to evaluate the antioxidant capacity of microalgal strains, and the contribution of carotenoids and phenols to the antioxidant activity of the extracts was also determined using multiple and unitary linear regression analysis. Therefore, these results provide an important basis for further enhancing the comprehensive development potential of microalgae strains, and are also expected to supply technical support for rapid evaluation of the comprehensive performance of microalgae biomass products in commercial development.

## 2. Materials and Methods

### 2.1. Strain and Culture Conditions

A total of 16 microalgae strains from different locations were used as experimental materials, and the detailed information on microalgae is given in Appendix A. The strains were cultured in 1500-mL vertical bubble column photobioreactors (6.0 cm × 60 cm) containing different mediums at 25 °C and continuous illumination. The medium for *Chlorella sorokiniana* SCSIO-46784, *Eustigmatos* sp. SCSIO-46716, *Scenedesmus* sp. SCSIO-46585, *Scenedesmus* sp. SCSIO-46579, *Scenedesmus* sp. SCSIO-46591, and *Uronema* sp. SCSIO-46782 was BG11 liquid culture medium (Appendix A) [32]. The culture for *Nannochloropsis* sp. SCSIO-45217, *Nannochloropsis* sp. SCSIO-45006, *Nannochloropsis* sp. SCSIO-45224, and *Phaeodactylum tricornutum* SCSIO-45120 was F/2 medium (Appendix A) [33]. *Porphyridium cruentum* SCSIO-45949 and *Rhodosorus* sp. SCSIO-45707 were cultured using ASW medium (Appendix A) [34]. ZNST medium was used for *Asterarcys* sp. SCSIO-46548 and *Asterarcys* sp. SCSIO-45829 (Appendix A), with the medium prepared using fresh water or seawater for each sample, respectively. *Euglena gracilis* was purchased from Freshwater Algae Culture of Hydrobiology Collection at the Institute of Hydrobiology, Chinese Academy of Sciences (Wuhan, China), strain No. FACHB-849, and we renumbered the algal strain as *Euglena gracilis* SCSIO-46781. The cultures used for *E. gracilis* SCSIO-46781 and *Arthrospira platensis* SCSIO-44012 were EGM (Appendix A) and Zarrouk (Appendix A) medium, respectively. The cells were harvested by centrifugation at 8000 rpm for 5 min under the later stage of the exponential/linear phase, then washed twice with sterile deionized water, freeze-dried (FD-1-50, Beijing Boyikang Laboratory Instrument Co., Ltd., Beijing, China), ground into powder, and stored at −20 °C for further extraction experiments.

### 2.2. Antioxidant Extracts Preparation

Two different solid–liquid extraction methods were adopted to prepare microalgae antioxidant extracts with the advantages of low cost, simple operation, no special equipment required, high yield, and excellent quality of the final product [35,36]. The extraction process was carried out in an inert nitrogen atmosphere in the dark at room temperature. The one-step extraction method was used to extract both apolar and polar compounds using an ethanol/water mixture. A 100-mg freeze-dried biomass was extracted using 2 mL of ethanol/water (3:1, *v*/*v*) and mixed for 30 min. The process was repeated twice, then the crude extract was centrifuged at 8000 rpm for 10 min, and the supernatant was collected and stored under a nitrogen atmosphere at −20 °C. The three-step extraction method aimed to separate apolar and polar compounds successively using hexane, ethylacetate, and hot water [37]. A 100-mg lyophilized algae powder was extracted using 2 mL of hexane for 30 min, then the extracts were centrifuged at 4500 rpm for 10 min to collect the supernatant. The process was repeated twice, then the algae residue was extracted using 2 mL of ethylacetate for 30 min and repeated twice. The supernatants were combined after centrifuging. Finally, the algae residue was extracted using 2 mL of hot water for 30 min in an 80 °C water bath and repeated twice. The supernatant was centrifuged at 4500 rpm for 10 min and collected. The hexane and ethylacetate extraction supernatants were dried using nitrogen, and 100% ethanol was added to complete the volume to 4 mL. The resulting mixture was stored with the water extraction supernatant in a nitrogen atmosphere at −20 °C.

### 2.3. Determination of Total Carotenoids Content

A 0.1-mL extraction solution was diluted with 90% methanol to 3 mL, and the absorbances of the mixture were obtained at 662, 645, and 470 nm using a TU-1810 UV-visible spectrophotometer (Persee Instrument Co., Ltd., Beijing, China). The total carotenoid content (TCC) was calculated according to the equations reported by Wang et al., (2022) [38].

### 2.4. Determination of Total Phenol Content

The Folin–Ciocalteu assay was adopted to determine the total phenol content of the extracts using gallic acid as a standard [6]. In brief, 10 μL of the extract was mixed with 95 μL of Folin–Ciocalteu reagent and kept at room temperature for 5 min. A total of 95 μL of sodium bicarbonate solution (60 g L^−1^) was added to the mixture and incubated at room temperature for 90 min. The absorbance at 750 nm was measured using a Biotek Epoch2 microplate spectrophotometer (Biotek Instruments, Inc., Winooski, VT, USA). Phenolic quantification was measured and expressed as mg per g of dry biomass based on the standard curve of gallic acid.

### 2.5. Antioxidant Activity Assay

#### 2.5.1. ABTS (2,2-azino-bis 3-ethylbenzthiazoline-6-sulphonic Acid) Radical-Scavenging Ability

The ABTS radical scavenging activity assay was performed as previously described, with some modifications [39]. Briefly, ABTS radical cation solution was prepared through 12–16 h reactions of ABTS (7.4 mM) with potassium persulfate (2.6 mM) at room temperature in the dark. The solution was diluted 20.3 times using H_2_O to obtain an absorbance of 0.700 at 734 nm. A total of 20 μL of extract or positive control (Trolox) was mixed with the diluted ABTS (180 μL). The mixture was shaken for 10 s and left undisturbed for 6 min. The absorbance at 734 nm was measured using a Biotek Epoch2 microplate spectrophotometer (Biotek Instruments, Inc., Vermont, America). The ABTS radical scavenging activity was calculated as follows:ABTS scavenging rate (%) = [1 − (A_sample_ − A_sample blank_)/A_control_] × 100(1)
where A_sample_ is the absorbance of ABTS solution and extracts, A_sample blank_ is the absorbance of extracts and H_2_O, and A_control_ is the absorbance of ABTS solution and H_2_O.

The ABTS radical scavenging capacity was expressed as μmol Trolox g^−1^ dry weight of microalgae.

#### 2.5.2. DPPH (2,2-Diphenyl-1-picrylhydrazyl) Radical Scavenging Ability

DPPH radical scavenging activity was measured using the method of Venkatesan et al., (2019) [40]. In brief, 180 μL of 0.1 mM DPPH methanolic solution was added to 20 μL of the extract solution. The solution was vortexed for 1 min and kept at room temperature for 30 min in the dark, and the absorbance at 517 nm was measured using a Biotek Epoch2 microplate spectrophotometer (Biotek Instruments, Inc., Winooski, VT, USA). Trolox was used as a positive control. The activity to scavenge DPPH radical was calculated using the following formula:DPPH scavenging rate (%) = [1 − (A_sample_ − A_sample blank_)/A_control_] × 100(2)
where A_sample_ is the DPPH solution and the extract sample, A_sample blank_ is the extract sample without DPPH solution, and A_control_ is the DPPH solution without the extract sample.

The DPPH radical scavenging activity was expressed as μmol Trolox g^−1^ dry weight of microalgae.

#### 2.5.3. Ferric Reducing Antioxidant Power (FRAP)

The reducing power of microalgae extracts were evaluated using the Fe^3+^–Fe^2+^ transformation method and the FRAP assay, which was modified as previously described [41]. In detail, The FRAP reagent was prepared from a mixture of 0.3 M acetate buffer (pH 3.6), 10 mM TPTZ, and 20 mM ferric chloride solution (10:1:1, *v*:*v*:*v*). A total of 195 μL of FRAP solution was mixed with 5 μL of extracts and incubated at room temperature for 15 min in the dark. The absorbance was measured at 593 nm using a microplate reader. Trolox was used as a positive control. The FRAP of the algae extracts was estimated and expressed as μmol Trolox g^−1^ of dry weight biomass.

### 2.6. Assessment of Overall Antioxidant Potential

A weighted scoring system was used to rapidly evaluate the overall antioxidant activity of the microalgal biomass [42]. For the ABTS, DPPH, and FRAP experiments, the lowest antioxidant activity score was recorded as “0”, the highest score was recorded as “1”, and the median value was calculated using linear approximation. The antioxidant test scores of the microalgae biomass were added together, and the sum was the overall antioxidant activity of the extracts.

### 2.7. Statistical Analysis

Data are presented as mean and standard deviations of triplicate experiments (*n* = 3) using a statistical system (origin 8.5). The statistical program IBM SPSS statistics 25 was used to analyze the data, and R Programming Language (v4.0.4) (TUNA Team, Tsinghua University, Beijing, China) was used for multiple regression analyses. *p* < 0.05 was considered statistically significant.

## 3. Results and Discussion

### 3.1. Carotenoid and Phenol Contents

Microalgae can produce various carotenoids, such as β-carotene, astaxanthin, fucoxanthin, lutein, canthaxanthin, zeaxanthin, lycopene and others [43]. Carotenoid content is an essential parameter for evaluating the antioxidant capacity of microalgae [13]. Different carotenoids express different polarities, solubilities, and chemical stabilities, which is why certain differences were found in the carotenoid content obtained using two different solid–liquid extraction methods [44]. Figure 1a shows the TCC of the ethanol/water extracts and the consecutive three-step fractionating procedure extracts of 16 species of microalgae. The TCC varied widely among the different microalgal strains, as well as when two different solid–liquid extraction procedures were used. In the one-step extraction of the investigated microalgal strains, *P. tricornutum* SCSIO-45120 had the highest TCC (5.45 mg g^−1^ biomass) and *Rhodosorus* sp. SCSIO-45707 had the lowest TCC (0.02 mg g^−1^ biomass). Additionally, in the three-step fractionating extraction procedure, the ethylacetate fraction contained the highest TCC, followed by the hexane fraction and the hot water fraction. The highest value of TCC was found in *E. gracilis* SCSIO-46781 (6.07 mg g^−1^ biomass) using the three-step fractionating extraction, which was significantly higher than when one-step extraction was used (Appendix A). In general, *E. gracilis* SCSIO-46781, *Nannochloropsis* sp. SCSIO-45224, and *A. platensis* SCSIO-44012 obtained higher TCC values (>1 mg g^−1^ biomass) via two different extraction procedures, while these two methods were not suitable for *Eustigmatos* sp. SCSIO-46716, *Rhodosorus* sp. SCSIO-45707, or *Asterarcys* sp. SCSIO-46548. On the other hand, some microalgal extracts had higher TCC using the one-step extraction than the consecutive three-step fractionating procedure, such as *P. tricornutum* SCSIO-45120 and *Uronema* sp. SCSIO-46782. Kim et al. found that ethanol provided the best fucoxanthin extraction yield for *P. tricornutum* [45].

Phenol compounds are widely found in land plants, and microalgae also contain phenol compounds, but little research has been reported among them [46]. It has been reported that phenol compounds have numerous bioactive properties, such as anti-tumor, antiviral, antimicrobial, and immunomodulatory effects, providing new potential resources for food and pharmaceutical products [47]. As shown in Figure 1b, the TPC varied in different strains of microalgae, which ranged from 0.43–3.90 and 0.86–11.73 mg g^−1^ of biomass using one-step extraction and three-step fractionating extraction, respectively. During the three-step fractionating extraction, it was common that the highest TPC value appeared in the hot water fraction, followed by the hexane and ethylacetate fraction, which was opposite to TCC. The TPC of *E. gracilis* SCSIO-46781 ranked first, regardless of which extraction method was used. On the other hand, the TPC in the *P. tricornutum* SCSIO-45120 extract was lower than that reported by Haoujar et al., (2019) [48] (39.39 mg g^−1^ biomass) and higher than that found by Goiris et al., (2012) [35] (1.40 mg g^−1^ biomass). It is worth noting that the polyphenolic composition and content can substantially vary as a function of microalgae growth conditions (nutrients availability, temperature, stress application) and the extracting solvents used for the evaluation of the antioxidant activity [48]. Generally, the TPC of different microalgae acquired suing the one-step extraction method was lower than that of the consecutive three-step fractionating procedure. For example, the TPC of *E. gracilis* SCSIO-46781 and *A. platensis* SCSIO-44012 obtained using the three-step method was more than three times than that of the one-step method (Appendix A). It had been proven that different phenolic compounds had specific selectivity to the kind of solvent used, which made the continuous three-step fractionation process more adequate for extract phenols from microalgae. Different phenols have different extraction methods due to their polarities. Currently, solid phase, solid–liquid, ultrasound-assisted, microwave-assisted, enzyme-assisted, and supercritical CO_2_ extraction methods have been developed [49].

### 3.2. Antioxidant Activities

There are more than 100 different assays that have been used to measure the antioxidant activity and free radical scavenging capacity of compounds [50]. Different compounds have different sensitivities to different assays, so there is no single antioxidant assay that can determine the total antioxidant capacity of all compounds in the extract [51]. Given the complex chemical compositions of microalgae extracts, we employed three widely used methods to determine their antioxidant capacity: ABTS, DPPH, and FRAP assays. These three methods of antioxidant activity evaluation have characteristics such as a clear principle and simple operation, reflecting the ability of substances to inhibit or remove most free radicals, as well as good biological relevance [52,53].

The ABTS assay is an important tool for evaluating the potential antioxidant capacity of extracts [54]. As shown in Figure 2a, the ABTS radical scavenging activity of the ethanol/water extracts from 16 microalgae strains varied between 3.67 and 29.60 μmol g^−1^ of biomass. *E. gracilis* SCSIO-46781 had the highest activity, corresponding to its high TCC and TPC in the ethanol/water fraction. Various fractions of the three-step extraction program exhibited different activities. The ABTS scavenging ability of the hexane fraction increased from 0.00 to 23.13 μmol g^−1^ of biomass, the ethylacetate fraction increased from 0.00 to 19.52 μmol g^−1^ of biomass, and the hot water fraction increased from 4.87 to 22.10 μmol g^−1^ of biomass (Appendix A). The total ABTS radical scavenging activity of the fractionating procedure extracts from 16 microalgae strains was within the range of 5.86 to 56.06 μmol g^−1^ of biomass, which was higher than the ethanol/water extracts. This result was inconsistent with the result of Goiris et al., (2012) [35]. *P. tricornutum* SCSIO-45120 had higher ABTS scavenging capacity in the ethanol/water fraction (22.62 μmol g^−1^ biomass) than in the water fraction (22.10 μmol g^−1^ biomass). However, the TCC in the water fraction of *P. tricornutum* SCSIO-45120 was low, and its TPC was lower than in the ethanol/water fraction (Figure 1). In other words, the microalgae extracts contained other antioxidants, which had vigorous ABTS scavenging activity.

The DPPH assay is another important tool for evaluating potential antioxidant capacity. Figure 2b illustrates the DPPH scavenging capacities of the ethanol/water extracts and the fractionating procedure extracts from 16 microalgae strains. As for the ethanol/water extracts, the values of the DPPH scavenging ability were within the range of 2.06–19.99 μmol g^−1^ of biomass, and the maximum scavenging effect was observed in *A. platensis* SCSIO-44012. However, DPPH radical scavenging activity was not detected in the hexane fractions of *Rhodosorus* sp. SCSIO-45707, *Scenedesmus* sp. SCSIO-46585, and *Scenedesmus* sp. SCSIO-46579, or in the ethylacetate fraction of *Rhodosorus* sp. SCSIO-45707. It was observed that the ethanol/water fractions of most microalgae had strong DPPH scavenging abilities. For instance, the maximum scavenging effect of *A. platensis* SCSIO-44012, *P. tricornutum* SCSIO-45120, *E. gracilis* SCSIO-46781, and *Nannochloropsis* sp. SCSIO-45224 were 19.99, 18.27, 16.84, and 14.20 μmol g^−1^ of biomass, respectively (Appendix A). The results of the DPPH scavenging activity corresponded to the ABTS scavenging assay. However, it must be noted that the water fraction of *P. tricornutum* SCSIO-45120 showed good ABTS scavenging activity (22.10 μmol g^−1^ biomass), while it had relatively low DPPH scavenging activity (3.20 μmol g^−1^ biomass). This result depended on the difference between the two free radical scavenging mechanisms [55].

In addition to these two methods, the FRAP assay has also been widely used to measure the antioxidant properties of plants and algae [56]. Figure 2c and Appendix A demonstrate the FRAP assay results of the ethanol/water extraction extracts and the fractionating procedure extracts from 16 microalgae strains. Their FRAP was within the range of 3.69–56.49 and 6.73–132.62 μmol g^−1^ of biomass, respectively. The highest FRAP was found for the *E. gracilis* SCSIO-46781 extracts, both via one-step or three-step extraction, as the extract of this strain contained the highest TPC, which was in accordance with previous results [57]. High FRAP (>40 μmol g^−1^ biomass) was also found in the ethanol/water and hexane fraction of *E. gracilis* SCSIO-46781 and the ethanol/water fraction of *P. tricornutum* SCSIO-45120. However, FRAP was not found in the hexane and ethylacetate extracts of *Rhodosorus* sp. SCSIO-45707, which was attributed to the small amounts of TCC and TPC contained within them. This indicates that TCC and TPC may have a strong correlation with the antioxidant activities of microalgae [58,59].

It is important to note that there is no single ideal test, and it is necessary to use several tests with different mechanisms of action to evaluate the entire antioxidant capacity of an extract or molecule [13]. The results of the antioxidant tests indicated that *E. gracilis* SCSIO-46781 exhibited the highest ABTS, DPPH, and FRAP activity. However, the values obtained using the three methods varied, which may be related to the different reaction mechanisms employed by each method. In general, ABTS radicals were easier to remove than DPPH radicals, which was consistent with our results [60].

### 3.3. Assessment of Overall Antioxidant Potential

The microalgae extracts contained different types of antioxidant substances, and it was difficult to evaluate the overall antioxidant activity of the extracts using a single antioxidant assay. Foo et al., (2017) evaluated the antioxidant activities of six species of algae by measuring ABTS radical scavenging ability and FRAP, and explored the correlation between TCC, TPC, and antioxidant activities using multiple regression analysis [41]. Hajimahmoodi et al., (2010) evaluated the antioxidant activities of the intracellular and extracellular substances of 12 microalgae strains using FRAP and DPPH-HPLC tests [61]. Esztella et al., (2017) used a weighted scoring system to evaluate the seasonal changes in the antioxidant activity of extracts from different leaves [62]. At present, there is a lack of comprehensive systematic evaluation of the overall antioxidant activity of microalgae biomass. The weighted scoring system has been commonly used to evaluate changes in the antioxidant activity of extracts [62]. Based on the diverse distinction of chemical contents and antioxidant activities observed between the two extraction methods discussed earlier, we made some modifications and adopted a score ranking system. By calculating the scores for both samples obtained using two extraction methods in the same system, we were able to evaluate the overall antioxidant potential of 16 microalgae biomass.

Table 1 shows that the five strains of microalgae with the highest antioxidant capacity scores were in the order of *E. gracilis* SCSIO-46781, *A. platensis* SCSIO-44012, *Nannochloropsis* sp. SCSIO-45224, *P. tricornutum* SCSIO-45120, and *Nannochloropsis* sp. SCSIO-45006. The high scores of these microalgae strains were mainly attributed to the three-step extraction method. Kottuparambil et al., (2019) reported that *E. gracilis* produced many antioxidants, such as β-carotene, L-ascorbic acid, α-tocopherol, wax esters, phytotoxins, and polyunsaturated fatty acids (PUFAs), which could be widely used in the manufacture of pharmaceuticals, cosmeceuticals, and nutraceuticals [63]. *Arthrospira platensis* (*Spirulina*) has been commercially cultivated on a large scale wordwide since the 1980s, and it is applied as a healthy food, feed, and medicine, forming the largest microalgae industry among the microalgae species [64]. In our study, the TPC and TCC of *A. platensis* SCSIO-44012 were not very high, but this strain presented a relatively marked antioxidant ability resulting from the phycocyanin extracted during the antioxidant extract preparation process. The literature has reported that water-soluble phycocyanin is the main component of *spirulina*, accounting for 15–25% of its dry biomass weight. It has been commercially produced as an antioxidant in several plants in China, Japan, and other countries [65]. *Nannochloropsis* is characterized by its high lipid content and carotenoid pigments, and it is recognized as containing potent natural antioxidants against the oxidation of unsaturated lipids. It can be utilized in nutraceutical applications and the development of functional products [66,67]. Interestingly, *P. tricornutum* SCSIO-45120 achieved a higher antioxidant score using the ethanol/water extraction compared to three-step extraction. This is because the TCC of *P. tricornutum* SCSIO-45120 in the ethanol/water extraction was much higher than that in the three-step extraction. Since the primary carotenoid of *P. tricornutum* was fucoxanthin, ethanol provided the best fucoxanthin extraction yield for this microalga [45]. It can be inferred that fucoxanthin might be an important component of the antioxidant compound in *P. tricornutum* SCSIO-45120, demonstrating its production potential [68]. In addition to the five strains, *C. sorokiniana* SCSIO-46784, *Uronema* sp. SCSIO-46782, and *Asterarcys* sp. SCSIO-45829 also had strong antioxidant activity. *C. sorokiniana* not only obtained the approval of “generally recognized as safe” from the US Drug and Food Administration due to its long history of human consumption and nutrient profile but was also one of the most suitable reservoirs of natural antioxidants [56]. It has been reported that *Uronema* has an average carbohydrate and protein content of 15.6% of its dry weight and 58% of its dry weight, respectively, which can theoretically be a good source for bioethanol and feed production [69,70]. Previous studies in our laboratory have found that *Asterarcys* sp. had a high yield of mycosporine-like amino acids, and this product had strong hydroxyl radical scavenging activity, which was close to that of the positive control (Ascorbic acid). These results indicate that this strain is a high-quality germplasm resource for the production of natural antioxidants [71].

The comprehensive antioxidant evaluation method was suitable for the complex evaluation of the antioxidant properties of plant extracts. To the best of our knowledge, this method has not been applied to evaluating the antioxidant properties of microalgae extracts. The weighted scoring system can quickly evaluate the overall antioxidant capacity of microalgae biomass. Overall, the weighted scoring system experiments demonstrated that among 16 microalgae strains, *E. gracilis* SCSIO-46781 exhibited the strongest antioxidant activity, guiding the development of this strain as a promising source of antioxidants.

### 3.4. Correlation Analysis between TCC, TPC, and Antioxidant Capacity of Microalgae Extracts

The changes in TCC and TPC significantly affected ABTS and DPPH radicals scavenging effects and FRAP. Multiple regression analysis of both TCC and TPC versus antioxidant activity are shown in Table 2. The R^2^ values indicated that phenols played a more important role in the FRAP assay of extracts than carotenoids. Almendinger et al., (2021) measured the TCC, TPC, and antioxidant activity of 13 microalgae strains, observing that the TPC had little correlation with their antioxidant activity [72]. Horincar et al., (2011) measured the antioxidant activity and the TPC of the extracts of five macroalgae strains, finding that phenolic compounds had significant contributions to the antioxidant capacity of the extracts [73]. Silva et al., (2021) measured the TPC, TCC, and radical scavenging ability of different solvent extracts from *Scenedesmus obliquus*, describing that the extracts with higher TCC or TPC had higher antioxidant activities [74]. Goiris et al., (2012) measured the TPC, TCC, and antioxidant activities of extracts of 32 microalgae strains using ABTS free radical scavenging ability, FRAP, and the linoleic acid auto-oxygen method [35]. It was found that the TCC and TPC were significantly correlated with the antioxidant capacity of the microalgae extracts. In addition, the R^2^ values of ABTS (0.67), DPPH (0.72), and FRAP (0.89) indicated that other substances other than carotenoids and phenols contributed to the antioxidant capacity of the extracts. Of note, microalgae could produce various antioxidant compounds, such as polysaccharides, polyunsaturated fatty acids, proteins, etc. [13]. Figure 3 describes the unitary linear regression between the overall antioxidant capacity, TCC, and TPC. Significant correlations were found between the antioxidative capacity scores and the TCC and TPC. The R^2^ of the TPC was higher than that of the TCC, indicating that phenols had a greater antioxidant capacity, which was similar to the results of the multiple regression analysis. Furthermore, in microalgae, the TPC increased upon exposure to UV-light, suggesting that it indeed played a role in the antioxidative response to stress [48].

The astaxanthin content of *H. pluvialis* was enhanced when the cells were exposed to nutrient deficiencies such as nitrogen, phosphorus, sulfur or salt stress, high temperatures, and strong light [75]. Kobayashi et al., (2003) [76] also found that the production of microalgae astaxanthin was enhanced due to an increase in reactive oxygen species (ROS), indicating that oxidative stress could be used as a culture method to upregulate the production of specific and desired antioxidant compounds in microalgae [77]. Additionally, extreme environments also offer the advantage of reducing potential contamination from competing microorganisms when cultivated under conditions of low pH, high alkali, high salinity, or nutritional limitation [78,79,80]. Microalgae can culture under unfavorable conditions for microorganisms or competing microalgae, which is of great importance to the large-scale (especially outdoor cultivation) and low-cost commercial production of algae. Therefore, microalgae with high antioxidant activity may be more suitable for large-scale outdoor cultivation.

## 4. Conclusions

This research established a rapid screening system for microalgae as a potential source of natural antioxidants. A positive correlation was proved between the antioxidant activity of microalgae extracts and their carotenoid and phenol content. *E. gracilis* SCSIO-46781, *A. platensis* SCSIO-44012, *Nannochloropsis* sp. SCSIO-45224, *P. tricornutum* SCSIO-45120, and *Nannochloropsis* sp. SCSIO-45006 extracts exhibited strong antioxidant capacity among the investigated 16 microalgae strains. Given that the above five microalgae strains have been widely used commercially, it can be concluded that the evaluation system not only screened for microalgae antioxidants, but also served as an effective model for screening excellent microalgae species with strong adaptability to environmental stress. The evaluation and screening system has broad application prospects in rapidly excavating new microalgae species/strains as feedstocks for nutraceuticals, pharmaceuticals, cosmeceuticals, and food additives. However, there is a strong correlation between the determination of the total antioxidant activity of microalgae biomass and extraction method used, so it is necessary to further explore extraction methods suitable for the evaluation of the antioxidant activity of microalgae. This will lay a foundation for the exploration of new antioxidant substances. Furthermore, various other antioxidant activity assay methods will be carried out to make this system more convincing, in addition to the three antioxidant activity assays of ABTS, DPPH, and FRAP.

## Figures and Tables

**Figure 1 foods-12-02652-f001:**
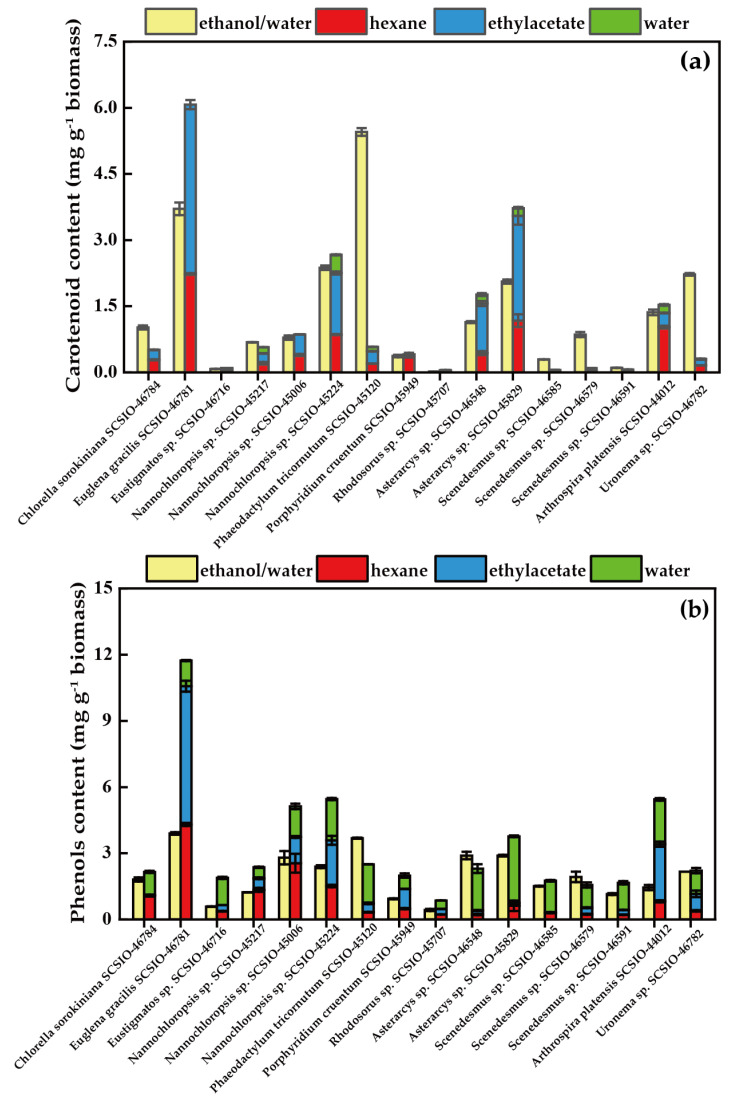
TCC (**a**) and TPC (**b**) of the ethanol/water extraction extracts and the fractionating procedure extracts for 16 microalgae strains.

**Figure 2 foods-12-02652-f002:**
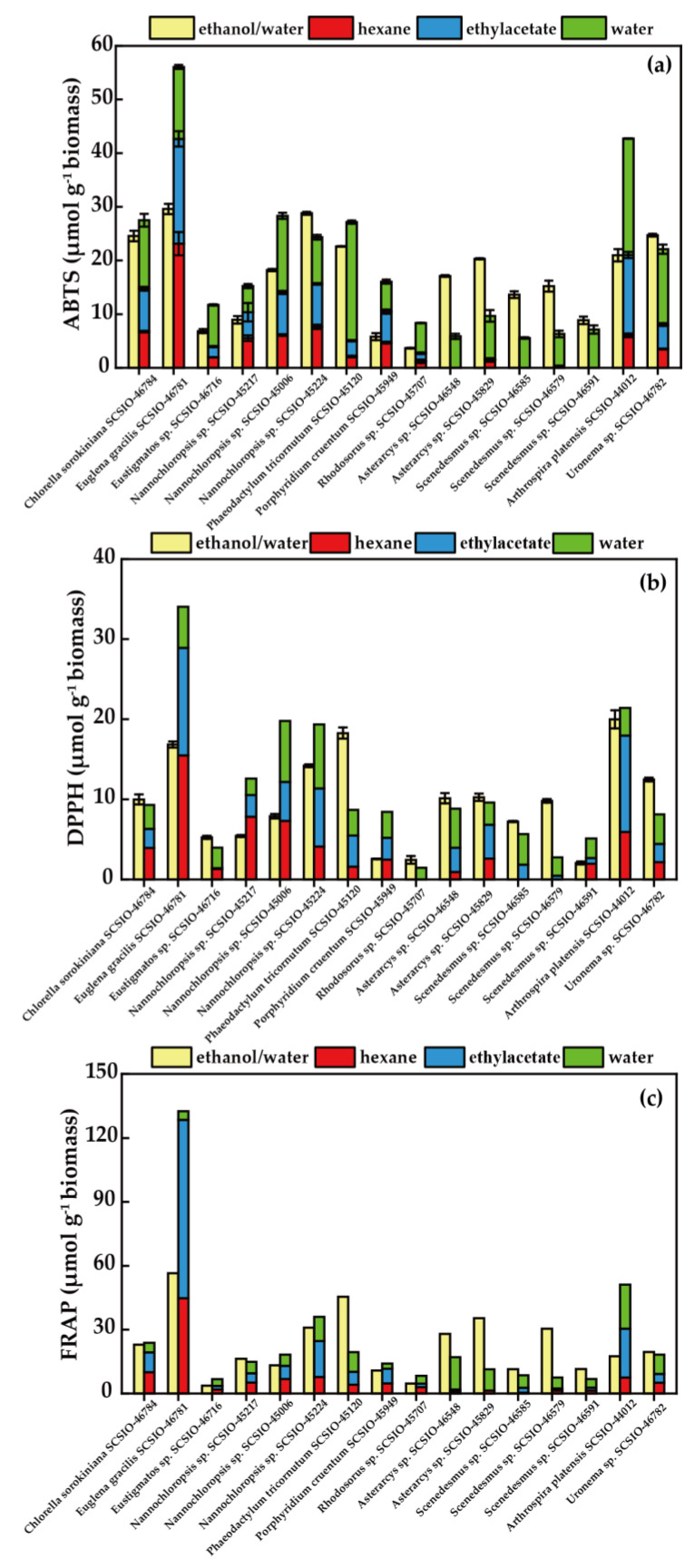
Antioxidant activities of the ethanol/water extraction extracts and the fractionating procedure extracts from 16 microalgae strains. Scavenging effects against ABTS radicals (**a**); DPPH radicals (**b**); FRAP (**c**).

**Figure 3 foods-12-02652-f003:**
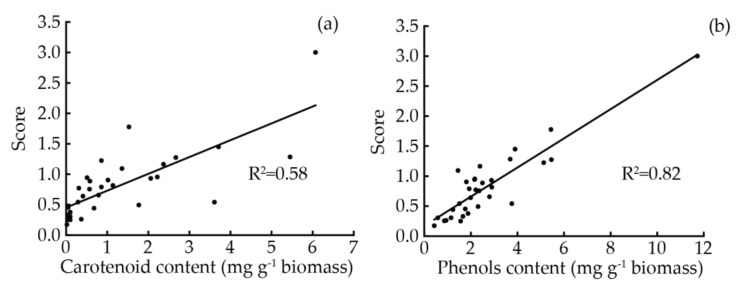
Correlation between antioxidant capacity scores and TCC (**a**) and TPC (**b**) for the microalgae extracts.

**Table 1 foods-12-02652-t001:** Scores of ABTS, DPPH, and FRAP assays and overall antioxidant capacity of one-step extracts and three-step extracts from 16 microalgae strains.

Species	ABTS	DPPH	FRAP	Total Score	Average Score	Rank
*Chlorella sorokiniana*	0.44	0.29	0.17	0.90	0.92	6
SCSIO-46784 ^a^						
*Chlorella sorokiniana*	0.49	0.27	0.18	0.94		
SCSIO-46784 ^b^						
*Euglena gracilis*	0.53	0.50	0.43	1.46	2.23	1
SCSIO-46781 ^a^						
*Euglena gracilis*	1.00	1.00	1.00	3.00		
SCSIO-46781 ^b^						
*Eustigmatos* sp.	0.12	0.16	0.03	0.31	0.35	14
SCSIO-46716 ^a^						
*Eustigmatos* sp.	0.21	0.12	0.05	0.38		
SCSIO-46716 ^b^						
*Nannochloropsis* sp.	0.16	0.16	0.12	0.44	0.60	10
SCSIO-45217 ^a^						
*Nannochloropsis* sp.	0.27	0.37	0.11	0.75		
SCSIO-45217 ^b^						
*Nannochloropsis* sp.	0.33	0.23	0.10	0.66	0.95	5
SCSIO-45006 ^a^						
*Nannochloropsis* sp.	0.51	0.58	0.14	1.23		
SCSIO-45006 ^b^						
*Nannochloropsis* sp.	0.51	0.42	0.23	1.16	1.22	3
SCSIO-45224 ^a^						
*Nannochloropsis* sp.	0.43	0.57	0.27	1.27		
SCSIO-45224 ^b^						
*Phaeodactylum tricornutum*	0.40	0.54	0.34	1.28	1.09	4
SCSIO-45120 ^a^						
*Phaeodactylum tricornutum*	0.48	0.26	0.15	0.89		
SCSIO-45120 ^b^						
*Porphyridium cruentum*	0.10	0.08	0.08	0.26	0.46	13
SCSIO-45949 ^a^						
*Porphyridium cruentum*	0.29	0.25	0.11	0.65		
SCSIO-45949 ^b^						
*Rhodosorus* sp.	0.07	0.07	0.04	0.18	0.22	16
SCSIO-45707 ^a^						
*Rhodosorus* sp.	0.15	0.04	0.06	0.25		
SCSIO-45707 ^b^						
*Asterarcys* sp.	0.31	0.30	0.21	0.82	0.66	9
SCSIO-46548 ^a^						
*Asterarcys* sp.	0.11	0.26	0.13	0.50		
SCSIO-46548 ^b^						
*Asterarcys* sp.	0.36	0.30	0.27	0.93	0.74	8
SCSIO-45829 ^a^						
*Asterarcys* sp.	0.17	0.28	0.09	0.54		
SCSIO-45829 ^b^						
*Scenedesmus* sp.	0.24	0.21	0.09	0.54	0.50	12
SCSIO-46585 ^a^						
*Scenedesmus* sp.	0.22	0.17	0.06	0.45		
SCSIO-46585 ^b^						
*Scenedesmus* sp.	0.27	0.29	0.23	0.79	0.52	11
SCSIO-46579 ^a^						
*Scenedesmus* sp.	0.11	0.08	0.06	0.25		
SCSIO-46579 ^b^						
*Scenedesmus* sp.	0.16	0.06	0.09	0.31	0.32	15
SCSIO-46591 ^a^						
*Scenedesmus* sp.	0.13	0.15	0.05	0.33		
SCSIO-46591 ^b^						
*Arthrospira platensis*	0.37	0.59	0.13	1.09	1.44	2
SCSIO-44012 ^a^						
*Arthrospira platensis*	0.76	0.63	0.39	1.78		
SCSIO-44012 ^b^						
*Uronema* sp.	0.44	0.37	0.15	0.96	0.87	7
SCSIO-46782 ^a^						
*Uronema* sp.	0.39	0.24	0.14	0.77		
SCSIO-46782 ^b^						

Note: ^a^ and ^b^ stand for one-step extracts and three-step extracts from 16 microalgae strains, respectively. For the ABTS, DPPH, and FRAP experiments, the lowest antioxidant activity score was recorded as “0”, the highest score was recorded as “1”, and the median value was calculated using linear approximation. The total score was obtained by adding together the antioxidant test scores of the microalgae biomass, and the sum was the overall antioxidant activity of the extracts. Based on the diverse distinction of chemical contents and antioxidant activities between the two extraction methods, we calculated the average scores for both samples from two extraction methods in the same system, through which we determined the score ranking and evaluated the overall antioxidant potential of 16 microalgae biomasses.

**Table 2 foods-12-02652-t002:** Multiple regression analysis of both TCC and TPC versus antioxidant activity using *t*-tests to determine significance.

X-Variable	Y-Variable	Coefficients	Standard Error	R^2^	*t* Stat	*p*-Value
carotenoid content	ABTS value	2.66	1.10	0.67	2.41	1.99 × 10^−2^
phenols content		3.33	0.98		3.41	1.33 × 10^−3^
carotenoid content	DPPH value	2.03	0.64	0.72	3.18	2.59 × 10^−3^
phenols content		1.93	0.56		3.42	1.30 × 10^−3^
carotenoid content	FRAP value	4.50	1.21	0.89	3.72	5.29 × 10^−4^
phenols content		8.56	1.07		8.01	2.14 × 10^−10^

## Data Availability

Data is contained within the article or Appendix A.

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
