# Peer review of "Rapid Screening of Microalgae as Potential Sources of Natural Antioxidants"

_foods, 2023, doi:10.3390/foods12142652_

Round 1

Reviewer 1 Report

Antioxidants are very important substances for the body, especially exogenous ones. The availability of antioxidants extracted from microalgae is very low. The main constituents with antioxidant activity in microalgae are phenolic compounds and carotenoids.

The article shows several methods with different yields for extracting antioxidants from microalgae. Taking into account the environment and sustainability. The idea of taking advantage of microalgae as a source of antioxidants seems to be interesting given the amount obtained with the different extraction methods and the 16 varieties of microalgae applied to the study.

The study presents a well-structured and coherent design.

Easy to read.

It seeks to show the commercial potential of microalgae.

On lines 154, 167 and 178, the equipment information must also include the city.

Author Response

July 3rd, 2023

Dear reviewers and editors,

Thank you for your letter and for the reviewers’ comments concerning our manuscript entitled “Rapid Screening of Microalgae as Potential Sources of Natural Antioxidants”. Those comments are all valuable and very helpful for revising and improving our paper, as well as the important guiding significance to our researchers. We have checked the comments carefully and have made revisions which we hope meet with approval. We have uploaded the revised submission and highlighted all the alterations in red. The point-by-point response to the comments is given below.

Thank you very much for kindly considering the manuscript.

Best wishes,

Dr.

Hualian Wu

South China Sea Institute of Oceanology, Chinese Academy of Sciences, No.164 Xingang West Road, Guangzhou 510301, China

Tel: +86 20 89023195

Fax: +86 20 89023195

Responses to the reviewers’ points

To reviewer #1:

Question 1 Antioxidants are very important substances for the body, especially exogenous ones. The availability of antioxidants extracted from microalgae is very low. The main constituents with antioxidant activity in microalgae are phenolic compounds and carotenoids. The article shows several methods with different yields for extracting antioxidants from microalgae. Taking into account the environment and sustainability. The idea of taking advantage of microalgae as a source of antioxidants seems to be interesting given the amount obtained with the different extraction methods and the 16 varieties of microalgae applied to the study. The study presents a well-structured and coherent design. Easy to read. It seeks to show the commercial potential of microalgae. On lines 154, 167 and 178, the equipment information must also include the city.

Response: Thanks for your kind recognition and suggestion. We have supplemented the city information of the equipment as shown on lines 166-169, 180-182 and 192-194 in the manuscript.

Reviewer 2 Report

The article is well-written and structured. Only a few things to correct or modify (see the attached file). 

Only a few corrections are needed

Author Response

July 3rd, 2023

Dear reviewers and editors,

Thank you for your letter and for the reviewers’ comments concerning our manuscript entitled “Rapid Screening of Microalgae as Potential Sources of Natural Antioxidants”. Those comments are all valuable and very helpful for revising and improving our paper, as well as the important guiding significance to our researchers. We have checked the comments carefully and have made revisions which we hope meet with approval. We have uploaded the revised submission and highlighted all the alterations in red. The point-by-point response to the comments is given below.

Thank you very much for kindly considering the manuscript.

Best wishes,

Dr.

Hualian Wu

South China Sea Institute of Oceanology, Chinese Academy of Sciences, No.164 Xingang West Road, Guangzhou 510301, China

Tel: +86 20 89023195

Fax: +86 20 89023195

Responses to the reviewers’ points

To reviewer #2:

Question 1 Lines 22-25: Please clarify this sentence.

Response: Thanks for your helpful suggestion, and sorry for failing to adequately clarify the sentence. We have rewritten this sentence accordingly as shown on lines 22-24.

Question 2 Line 90: Correct "stains".

Response: Thanks for the meticulous checking. “stains” have been revised as “strains”. We are very sorry for this mistake.

Question 3 Lines 165-166: Please correct this sentence.

Response: Thanks for your kind suggestion. This sentence has been revised accordingly as shown on lines 180-182.

Question 4 Figures: I think it would be better to make color figures to make the histograms easier to read.

Response: Thanks for your comment. We have revised Figure 1 as the color figure to make it easier to read.

Question 5 Figures: Color histograms are more readable.

Response: Thanks for your comment. We have revised Figure 2 as the color figure to make it easier to read.

Question 6 Line 335: Please add "platensis" to make understand the subject A. platensis of the following sentence.

Response: Thanks for your kind suggestion. "platensis" has been added to the sentence in our revised manuscript as shown on lines 369-.372.

Question 7 Line 362: Please explain what is Vc.

Response: Thanks for the helpful mention, and sorry for not explaining the meaning of Vc clearly. We have modified the manuscript accordingly as shown on lines 394-398.

Question 8Table 1: RANK: It would be better to repeat in the caption where the "Rank" comes from and how it is calculated.

Response: Thanks for your kind suggestion. We have supplemented the note of Table 1 in the manuscript to explain where the "Rank" comes from and how it is calculated, which is shown on lines 4100-419.

For the ABTS, DPPH and FRAP experiments, the lowest antioxidant activity score was recorded as "0", the highest score was recorded as "1", and the median value was calculated using linear approximation. The total score was obtained by adding together the antioxidant test scores of microalgae biomass and the sum was the overall antioxidant activity of the extracts. Based on the diverse distinction of chemical contents and antioxidant activities between the two extraction methods, we calculated average scores for both samples from two extraction methods in the same system, through which we determined the score ranking and evaluated the overall antioxidant potential of 16 microalgae biomass.

Question 9Table 1: "Average score": In some cases the total values of extractions a and b are very different: does it make sense to calculate the average?

Response: Thanks for your constructive suggestion. There were a variety of substances with antioxidant activity in microalgae, and the chemical contents and antioxidant activities of the extracts obtained by different extraction methods had great differences. Therefore, we established a rapid evaluation system for the overall antioxidant activity of microalgae by calculating the average scores of two samples from two extraction methods in the same system, which was generally more reliable and convincing.

Question 10TEAC???

Response: Thanks for the meticulous checking. “TEAC” has been revised as “ABTS”. We are very sorry for this mistake.

Question 11 Comments on the quality of English language: Only a few corrections are needed.

Response: Thanks for the suggestion. We have revised the whole manuscript carefully and tried to avoid any grammar or syntax errors. In addition, we have asked several colleagues who are skilled authors of English language papers to check the English. We believe that the language is now acceptable for the review process.

Reviewer 3 Report

This manuscript is about rapid screening of microalgae as potential sources of natural antioxidants. It is interesting and I think, after complete revision of manuscript. You can find my comments in below:

1. The manuscript must be revised grammatically and the English level of it must be improved by a native editor.

2. The authors must re-write the abstract and conclusion sections.

3. In lines 43 to 45, it is better to first describe the antioxidants and various types of them with an appropriate reference.

4. In line 57, write some products about the use of antioxidants in those three industries.

5. In line 80, what are the extraction methods of microalgae biomass.

6. In line 85, please add more references beside references number 7 and 23.

7. In line 124, why did the author select solid-liquid extraction method? Please write the reasons for example because of higher yield.

8. In line 157, why only ABTS, FRAP and DPPH methods were selected? We have other methods for evaluation of antioxidants.

9. In line 252, give more references about the researches done on phenol compounds.

10. In antioxidant activities, please give more references about the methods. Also, it is better to add some sentences about comparison between these methods (ABTS, DPPH and FRAP).

11. If it is possible, please decrease the number of tables in the manuscript. There are a lot of tables and this could hesitate the reader.

12. Please increase the DPI value of figures. The DPI values of figures are low and the quality of them is poor.  

The manuscript must be revised grammatically and the English level of it must be improved by a native editor.

Author Response

July 3rd, 2023

Dear reviewers and editors,

Thank you for your letter and for the reviewers’ comments concerning our manuscript entitled “Rapid Screening of Microalgae as Potential Sources of Natural Antioxidants”. Those comments are all valuable and very helpful for revising and improving our paper, as well as the important guiding significance to our researchers. We have checked the comments carefully and have made revisions which we hope meet with approval. We have uploaded the revised submission and highlighted all the alterations in red. The point-by-point response to the comments is given below.

Thank you very much for kindly considering the manuscript.

Best wishes,

Dr.

Hualian Wu

South China Sea Institute of Oceanology, Chinese Academy of Sciences, No.164 Xingang West Road, Guangzhou 510301, China

Tel: +86 20 89023195

Fax: +86 20 89023195

Responses to the reviewers’ points

To reviewer #3:

Question 1 The manuscript must be revised grammatically and the English level of it must be improved by a native editor.

Response: Thanks for the suggestion. We have revised the whole manuscript carefully and tried to avoid any grammar or syntax errors. In addition, we have asked several colleagues who are skilled authors of English language papers to check the English. We believe that the language is now acceptable for the review process.

Question 2 The authors must re-write the abstract and conclusion sections.

Response: Thanks for your kind suggestion. We have rewritten the section of the abstract and conclusion in the manuscript accordingly.

Question 3 In lines 43 to 45, it is better to first describe the antioxidants and various types of them with an appropriate reference.

Response: Thanks for your kind suggestion, and we have modified the manuscript accordingly as shown on lines 41-50.

Question 4 In line 57, write some products about the use of antioxidants in those three industries.

Response: Thanks for your kind suggestion. We have written some products about the use of antioxidants in nutritional food, pharmaceutical and cosmetical industries in the manuscript as shown on lines 60-71.

There is a trend toward using natural bioactive compounds of microalgae as antioxidants in various fields, including the food, pharmaceutical, and cosmetic industries. The blend of inorganic nanoparticles and natural antioxidant chemicals of microalgae was popular in active food packaging due to their robust antibacterial, antioxidant, UV barrier, oxygen removal, and low environmental impact properties [1, 2]. Additionally, the food industry is applying whole microalgal biomass or their extracted purified compounds as novel ingredients for the formulation of food products such as baked goods, pasta, noodles, plant-based milk, soups and many others [3]. Adnan et al. (2020) [4] reported that beta-tocopherol and alpha-carotene from microalgae played the role of oxidants in prostate cancer by reducing cell viability and increasing ROS production and lipid peroxidation. Foo et al. (2021) [5] found that carotenoids could be used as a natural antioxidant in anti-aging skin care products in the cosmetics industry, in addition to being incorporated as a natural pigment to color cosmetics.

Question 5 In line 80, what are the extraction methods of microalgae biomass.

Response: Thanks for your question. Microalgae biomass should undergo an extraction procedure before use to make it a more valuable source of antioxidants [6]. Microalgae extracts typically contain a variety of bioactive ingredients [7]. There are various methods available for the extraction of microalgae antioxidant extracts, including solid-liquid, hot water, dilute acid, dilute alkali, enzyme, microwave, ultrasonic and supercritical fluid extraction methods [8]. Different extraction methods can affect the yield, composition, structure, and integrity of bioactive extracts.

Question 6 In line 85, please add more references beside references number 7 and 23.

Response: Thanks for your kind suggestion. We have added more references in the manuscript accordingly as shown on lines 98-100 to make the argument more convincing.

Question 7 In line 124, why did the author select solid-liquid extraction method? Please write the reasons for example because of higher yield.

Response: Thanks for your kind suggestion. The selection of a suitable extraction method is a crucial step in the extraction of microalgae antioxidant extracts from raw materials, because this decision may affect the yield, composition, structure, and integrity of bioactive extracts. There are various methods available for the extraction of microalgae antioxidant extracts, including solid-liquid, hot water, dilute acid, dilute alkali, enzyme, microwave, ultrasonic and supercritical fluid extraction methods [8]. Generally, solid-liquid extraction is the most common method for extracting microalgae antioxidants with the advantages of low cost, simple operation, no special equipment required, high yield and excellent quality of final product [9]. In addition, solid-liquid extraction is an important technology from an industrial point of view. We have modified the manuscript accordingly as shown on lines 138-140. In the future, we will plan to explore various advanced extraction techniques, such as microwave-assisted, ultrasonic-assisted, enzyme-assisted extraction and so on, promoting the large industrial preparation of microalgae antioxidant extracts.

Question 8 In line 157, why only ABTS, FRAP and DPPH methods were selected? We have other methods for evaluation of antioxidants.

Response: Thanks for your comment. We have revised the manuscript accordingly as shown on lines 278-286, 287-288, 308, 325-326 and 338-344.

There are over 100 different assays have been used to measure the antioxidant activity and free radical scavenging capacity of compounds [10]. Different compounds have different sensitivities to different assays, so there is no single antioxidant assay that can determine the total antioxidant capacity of all compounds in the extract [11]. Given the complex chemical compositions of microalgae extracts, we employed three widely used methods to determine their antioxidant capacities, such as ABTS, DPPH and FRAP assays. These three methods of antioxidant activity evaluation have the following characteristics, such as clear principle, simple operation; reflecting the ability of substances to inhibit or remove most free radicals and good biological relevance and so on [12, 13]. To get a more comprehensive overview of the antioxidant profile of the 16 algae samples investigated, we will plan to adopt other methods for measuring the antioxidant activity of microalgae extracts, such as hydroxyl radical scavenging, superoxide free radicals scavenging and ferrous irons chelating activity assay, et al.[14-16].

Question 9 In line 252, give more references about the research done on phenol compounds.

Response: Thanks for your kind suggestion. We have added more references about the research done on phenol compounds in the manuscript accordingly as shown on lines 262-267 and 444-445.

Question 10 In antioxidant activities, please give more references about the methods. Also, it is better to add some sentences about comparisons between these methods (ABTS, DPPH and FRAP).

Response: Thanks for your kind suggestion. We have revised the manuscript accordingly as shown on lines 278-286 and 338-344.

It is important to note that there is no single ideal test, and it is necessary to use several tests with different mechanisms of action to evaluate the whole antioxidant capacity of an extract or molecule [17]. The results of the antioxidant tests clearly indicate that E. gracilis SCSIO-46781 exhibited the highest ABTS, DPPH and FRAP activity, but their values were different, which may be related to these three methods with different reaction mechanisms. In general, ABTS radicals were generally easier to remove than DPPH radicals, which was consistent with our results [18].

Question 11 If it is possible, please decrease the number of tables in the manuscript. There are a lot of tables and this could hesitate the reader.

Response: Thanks for your kind suggestion. To attract more readers, we have tried our best to limit the number of tables in the manuscript, and the other tables have been added to the supplementary material. Tables S12-S16 in the supplementary material are intended to give the readers a clearer understanding of the specific data in Figures 1-2 in the manuscript.

Question 12 Please increase the DPI value of figures. The DPI values of figures are low and the quality of them is poor.

Response: Thanks for your kind suggestion. We have revised Figures 1 and 2 as the color figures and increased their DPI values to 300 ppi to make them easier to read.

Question 13 Comments on the quality of English language: The manuscript must be revised grammatically and the English level of it must be improved by a native editor.

Response: Thanks for your comment. As the review suggested we have revised the whole manuscript carefully and tried to avoid any grammar or syntax errors. In addition, we have asked several colleagues who are skilled authors of English language papers to check the English. We believe that the language is now acceptable for the review process.

References

  1. Yang, N.; Zhang, Q.; Chen, J.; Wu, S.; Chen, R.; Yao, L.; Li B.; Liu X.; Zhang R.; Zhang, Z. Study on Bioactive Compounds of Microalgae as Antioxidants in a Bibliometric Analysis and Visualization Perspective. Plant Sci. 2023, 14, 1144326.
  2. Vieira, I.R.S.; de Carvalho, A.P.A.D.; Conte‐Junior, C.A. Recent Advances in Biobased and Biodegradable Polymer nanocomposites, Nanoparticles, and Natural Antioxidants for Antibacterial and Antioxidant Food Packaging Applications. Rev. Food Sci. F. 2022, 21(4), 3673-3716.
  3. Ampofo J.; Abbey L. Microalgae: Bioactive Composition, Health Benefits, Safety and Prospects as Potential High-value Ingredients for the Functional Food Industry. Foods. 2022, 11(12), 1744.
  4. Adnan, A. Y. N. A. Apoptotic Effects of Beta-Carotene, Alpha-Tocopherol and Ascorbic Acid on PC-3 Prostate Cancer Cells. J. of Biol. and Chem. 2020, 48(3), 211-218.
  5. Foo, S.C.; Khoo, K.S.; Ooi, C.W.; Show, P.L.; Khong, N.M.; Yusoff, F.M. Meeting Sustainable Development Goals: Alternative Extraction Processes for Fucoxanthin in Algae. Bioeng. Biotech. 2021, 8, 546067.
  6. Cotas, J.; Leandro, A.; Monteiro, P.; Pacheco, D.; Figueirinha, A.; Gonçalves, A.M.; da Silva G.J.; Pereira, L. Seaweed Phenolics: From Extraction to Applications. drugs. 2020, 18(8), 384.
  7. Michalak, I.; Tiwari, R.; Dhawan, M.; Alagawany, M.; Farag, M.R.; Sharun, K.; Emran, T.B.; Dhama, K. Antioxidant Effects of Seaweeds and their Active Compounds on Animal Health and Production– A Review. Quart. 2022, 42(1), 48-67.
  8. Cong-Cong, X.U.; Bing, W.A.N.G.; Yi-Qiong, P.U.; Jian-Sheng, T.A.O.; Zhang, T. Advances in Extraction and Analysis of Phenolic Compounds from Plant Materials. J. Nat. Medicines. 2017, 15(10), 721-731.
  9. Gadkari, P.V.; Kadimi, U.S.; Balaraman, M. Catechin Concentrates of Garden Tea Leaves (Camellia sinensis L.): Extraction/Isolation and Evaluation of Chemical Composition. Sci. Food Agr. 2014, 94(14), 2921-2928.
  10. Sueishi, Y.; Nii, R.; Uda, C.; Takashima, A. Antioxidant Capacities in Various Animal Sera as Measured with Multiple Free-radical Scavenging Method. Med. Chem. Lett. 2019, 29(16), 2145-2149.
  11. Gündüz, M.; Çiçek, Ş.K.; Topuz, S. Extraction and Optimization of Phenolic Compounds from Butterbur Plant (Petasites hybridus) by Ultrasound-Assisted Extraction and Determination of Antioxidant and Antimicrobial Activity of Butterbur Extracts. Appl. Res. Med. Aroma. 2023, 100491.
  12. Mendonça, J.D.S.; Guimarães, R.D.C.A.; Zorgetto-Pinheiro, V.A.; Fernandes, C.D.P.; Marcelino, G.; Bogo, D.; Freitas, K.C.; Hiane, P.A.; de Padua Melo, E.S.; Vilela, M.L.B.; Nascimento, V.A.D. Natural Antioxidant Evaluation: A Review of Detection Methods. Molecules. 2022, 27(11), 3563.
  13. Flieger, J.; Flieger, W.; Baj, J.; Maciejewski, R. Antioxidants: Classification, Natural Sources, Activity/Capacity Measurements, and Usefulness for the Synthesis of Nanoparticles. Materials. 2021, 14(15), 4135.
  14. Wang, N.; Dai, L.M.; Chen, Z.S.; Li, T.; Wu, J.Y.; Wu, H.B.; Wu, H.L.; Xiang, W.Z. Extraction Optimization, Phys-icochemical Characterization, and Antioxidant Activity of Polysaccharides from Rhodosorus SCSIO-45730. J. Appl. Phycol. 2021, 34(1), 285-299.
  15. Wang, N.; Chen, Z.S.; Lv, J.T.; Li, T.; Wu, H.L.; Wu, J.Y.; Wu, H.B.; Xiang, W.Z. Characterization, Hypoglycemia and Antioxidant Activities of Polysaccharides from Rhodosorus sp. SCSIO-45730. Crop. Prod. 2023, 191, 115936.
  16. Haoujar, I.; Cacciola, F.; Abrini, J.; Mangraviti, D.; Giuffrida, D.; Oulad El Majdoub, Y.; Kounnoun, A.; Miceli, N.; Taviano, M.F.; Mondello, L.; Rigano, F.; Senhaji, S.N. The Contribution of Carotenoids, Phenolic Compounds, and Flavonoids to the Antioxidative Properties of Marine Microalgae Isolated from Mediterranean Morocco. Molecules. 2019, 24(22), 4037.
  17. Coulombier, N.; Jauffrais, T.; Lebouvier, N. Antioxidant Compounds from Microalgae: A Review. Drugs. 2021, 19(10), 549.
  18. Echegaray, N.; Pateiro, M.; Munekata, P.E.; Lorenzo, J.M.; Chabani, Z.; Farag, M.A.; Domínguez, R. Measurement of Antioxidant Capacity of Meat and Meat Products: Methods and Applications. Molecules. 2021, 26(13), 3880.
